# Non-parametric Inference Adaptive to Intrinsic Dimension

## Abstract

We consider non-parametric estimation and inference of conditional moment models in high dimensions. We show that even when the dimension $D$ of the conditioning variable is larger than the sample size $n$, estimation and inference is feasible as long as the distribution of the conditioning variable has small intrinsic dimension $d$, as measured by locally low doubling measures. Our estimation is based on a sub-sampled ensemble of the $k$-nearest neighbors ($k$-NN) $Z$-estimator. We show that if the intrinsic dimension of the covariate distribution is equal to $d$, then the finite sample estimation error of our estimator is of order $n^{-1/(d+2)}$ and our estimate is $n^{1/(d+2)}$-asymptotically normal, irrespective of $D$. The sub-sampling size required for achieving these results depends on the unknown intrinsic dimension $d$. We propose an adaptive data-driven approach for choosing this parameter and prove that it achieves the desired rates. We discuss extensions and applications to heterogeneous treatment effect estimation.

**Keywords:** non-parametric statistics, inference, intrinsic dimension, conditional moment equation

## 1. Introduction

Many non-parametric estimation problems in econometrics and causal inference can be formulated as finding a parameter vector $\theta(x) \in \mathbb{R}^p$ that is a solution to a set of conditional moment equations:

$$\mathbf{E}[\psi(Z; \theta(x))|X = x] = 0, \tag{1}$$

when given $n$ i.i.d. samples $(Z_1, \ldots, Z_n)$ from the distribution of $Z$, where $\psi : \mathcal{Z} \times \mathbb{R}^p \to \mathbb{R}^p$ is a known vector valued moment function, $\mathcal{Z}$ is an arbitrary data space, $X \in \mathcal{X} \subset \mathbb{R}^D$ is the feature vector that is included $Z$. Examples include non-parametric regression[1], quantile regression[2], heterogeneous treatment effect estimation[3], instrumental variable regression[4], local maximum likelihood estimation[5] and estimation of structural econometric models (see e.g., Reiss and Wolak (2007) and examples in Chernozhukov et al. (2016); Chernozhukov et al. (2018b)). The study of such conditional moment restriction problems has a long history in econometrics (see e.g., Newey (1993); Ai and Chen (2003); Chen and Pouzo (2009); Chernozhukov et al. (2015); Chen et al. (2016)). However, the majority of the literature assumes that the conditioning variable $X$ is low dimensional, i.e. $D$ is a constant as the sample size $n$ grows (see e.g., Athey et al. (2019)).[6]

---

1. $Z = (X, Y)$, where $Y \in \mathbb{R}^p$ is the dependent variable, and $\psi(Z; \theta(x)) = Y - \theta(x)$.
2. $Z = (X, Y)$ and $\psi(Z; \theta(x)) = 1\{Y \leq \theta(x)\} - \alpha$, for some $\alpha \in [0, 1]$ that denotes the target quantile.
3. $Z = (X, T, Y)$, where $T \in \mathbb{R}^p$ is a vector of treatments, and $\psi(Z; \theta(x)) = (Y - \langle \theta(x), T \rangle) T$.
4. $Z = (X, T, W, Y)$, where $T \in \mathbb{R}$ is a treatment, $W \in \mathbb{R}$ an instrument and $\psi(Z; \theta(x)) = (Y - \theta(x) T) W$.
5. Where the distribution of $Z$ admits a known density $f(z; \theta(x))$ and $\psi(Z; \theta(x)) = \nabla_\theta \log(f(Z; \theta(x)))$.
6. Notable exceptions include high dimensional models under parametric assumptions on $\theta(x)$, such as sparse linear forms (see e.g., Chernozhukov et al. (2018a)). There is also work that addresses the fully non-parametric setup (see e.g., Lafferty and Wasserman (2008); Dasgupta and Freund (2008); Kpotufe (2011); Biau (2012); Scornet et al. (2015)) but those are focused on the estimation problem, and do not address inference (i.e., constructing asymptotically valid confidence intervals).

Recent studies demonstrate the success of non-parametric methods (see e.g., Lewis and Syrgkanis (2018)) for solving conditional moment equations even in the high-dimensional settings. Yet, there are limited theoretical results that explain why these methods work well. Indeed without any further structural assumptions on the problem, the exponential in dimension rates of approximately $n^{1/D}$ (see e.g., Stone (1982)) cannot be avoided. Thereby estimation is in-feasible even if $D$ grows very slowly with $n$.

One hypothesis is that the *intrinsic dimension* of the conditioning variables is low (i.e. even though $X$ is high dimensional, its coordinates are highly correlated), and that causal machine learning estimators are adaptive to this hidden low dimensional structure in the data.[7] *Our work makes this argument, establishing estimation and asymptotic normality results for the general conditional moment problem, with rates that only depend on the intrinsic dimension, independent of the explicit dimension of the conditioning variable.*

We build on two literatures. The statistical machine learning literature introduces the notion of intrinsic dimension, which is defined by saying that the distribution of $X$ has a small doubling measure around the target point $x$. Under assumptions of low intrinsic dimension papers in this literature establish fast estimation rates in high-dimensional kernel regression settings (Dasgupta and Freund, 2008; Kpotufe, 2011; Kpotufe and Garg, 2013; Xue and Kpotufe, 2018; Chen and Shah, 2018; Kim et al., 2018; Jiang, 2017). However these results do not apply to the conditional moment problems we study here. In the econometrics literature, the pioneering work of Wager and Athey (2018); Athey et al. (2019) does address estimation and inference of conditional moment models, but only in the low dimensional regime.[8] Relative to these literatures, our contributions are as follows:

- We extend the asymptotic normality results of Wager and Athey (2018); Athey et al. (2019) to general sub-sampled kernel estimators and for vector valued parameters $\theta(x)$. Our analysis also allows us to establish rates in the high-dimensional low intrinsic dimension regime. Given samples $S = (Z_1, \ldots, Z_n)$, our estimator solves a locally weighted empirical conditional moment equation

$$\hat{\theta}(x) \text{ solves} : \sum_{i=1}^{n} K(x, X_i, S) \, \psi(Z_i; \theta) = 0 \,, \tag{2}$$

  where $K(x, X_i, S)$ is a *kernel* capturing the proximity of $X_i$ to the target point $x$. We consider weights $K(x, X_i, S)$ that take the form of an average over $B$ base weights: $K(x, X_i, S) = \frac{1}{B} \sum_{b=1}^{B} K(x, X_i, S_b) \, 1\{i \in S_b\}$, where each $K(x, X_i, S_b)$ is calculated based on a randomly drawn sub-sample $S_b$ of size $s < n$ from the original sample.

- Our main estimation and asymptotic normality results (see Theorems 6 and 7), are stated in terms of two high-level quantities, specifically an upper bound $\epsilon(s)$ on the rate at which the

---

7. This observation builds on a long line of work in machine learning (Dasgupta and Freund, 2008; Kpotufe, 2011; Kpotufe and Garg, 2013).

8. These results have been extended in multiple directions, such as improved rates through local linear smoothing Friedberg et al. (2018), robustness to nuisance parameter estimation error Oprescu et al. (2018) and improved bias analysis via sub-sampled nearest neighbor estimation Fan et al. (2018). However, they all require low dimensional setting and the rate provided by the theoretical analysis is roughly $n^{-1/D}$, translating to $\Omega(\epsilon^{-D})$ samples for getting a confidence interval of size $\epsilon$, which is prohibitive in most target applications of machine learning based econometrics. In particular, Wager and Athey (2018) consider regression and heterogeneous treatment effect estimation with a scalar $\theta(x)$ and prove $n^{1/D}$-asymptotic normality of a sub-sampled random forest based estimator and Athey et al. (2019) extend it to the general conditional moment settings.

kernel "shrinks" and a lower bound $\eta(s)$ on the "incrementality" of the kernel. Notably, the explicit dimension of the conditioning variable $D$ does not enter the theorem, so it suffices in what follows to show that $\epsilon(s)$ and $\eta(s)$ depend only on $d$. The shrinkage rate $\epsilon(s)$ is defined as the $\ell_2$-distance between the target point $x$ and the farthest point on which the kernel places positive weight $X_i$, when trained on a data set of $s$ samplesIncrementality of a kernel describes how much information is revealed about the weight of a sample $i$ solely by knowledge of $X_i$, and is captured by the second moment of the conditional expected weight The sub-sampling size $s$ can be used to control both shrinkage and incrementality and for trading-off correctly between bias and variance. We also prove that incrementality can be lower bounded as a function of kernel shrinkage, so that having a sufficiently low shrinkage rate enables both estimation and inference. Corollary 11 and Lemma 13), rather than the explicit dimension $D$. In particular, we show that $\epsilon(s) = O(s^{-1/d})$ and $\eta(s) = \Theta(1/s)$, which lead to our main theorem *that the sub-sampled $k$-NN estimate achieves an estimation rate of order $n^{1/(d+2)}$ and is also $n^{1/(d+2)}$-asymptotically normal (Theorems 12 and 15).*

- We provide a closed form characterization of the asymptotic variance of the sub-sampled $k$-NN estimate, based on the conditional variance moments defined as $\sigma^2(x) = \text{Var}\left(\psi(Z;\theta) \mid X = x\right)$ (Theorem 14 and Eq. (14)). For example, for the 1-NN kernel, the asymptotic variance is $\text{Var}(\hat{\theta}(x)) = \frac{\sigma^2(x)s^2}{n(2s-1)}$. This strengthens prior results of Fan et al. (2018) and Wager and Athey (2018), which only proved the existence of an asymptotic variance without providing an explicit form (and thereby relied on bootstrap approaches for the construction of confidence intervals). Our Monte Carlo study shows that our constructed confidence intervals provide great finite sample coverage in a high dimensional regression setup (see Figure 1)[9].

- The sub-sampling size required to achieve optimal rates depends on the intrinsic dimension which is unknown. We discuss an adaptive data-driven approach for picking the sub-sample size $s$ so as to achieve near-optimal estimation or asymptotic normality rates, adapting to the unknown intrinsic dimension of data (see Propositions 16 and 17). Figure 2 depicts the performance of our adaptive approach compared to two benchmarks, one constructed based on theory for intrinsic dimension $d$ which may be unknown, and the other one constructed naïvely based on the known but sub-optimal extrinsic dimension $D$. As can be observed, our adaptive approach selects $s$ close to the value suggested by the theory and therefore leads to a compelling finite sample coverage[10].

**Structure of the paper.** The rest of the paper is organized as follows. In §2, we provide preliminary definitions, in §2.1 and §2.2 we explain our algorithms, in §2.3 we explain doubling dimension (see Appendix B for examples). In §3 we state our assumptions, in §4 we provide general estimation and inference results for kernels that satisfy shrinkage and incrementality conditions, and in §5 we apply such results to the $k$-NN kernel and prove estimation and inference rates for such kernels that only depend on intrinsic dimension. We discuss the extension to heterogeneous treatment effect estimation in §6 and and defer technical proofs to Appendices.

## 2. Preliminaries

Suppose we have a data set $M$ of $n$ observations $Z_1, Z_2, \ldots, Z_n$ drawn independently from some distribution $\mathcal{D}$ over the observation domain $\mathcal{Z}$. We focus on the case that $Z_i = (X_i, Y_i)$, where $X_i$

---

9. See Appendix C for detailed explanation of our simulations
10. Code is available via https://anonymous.4open.science/r/inference-intrinsic-dimension-E037

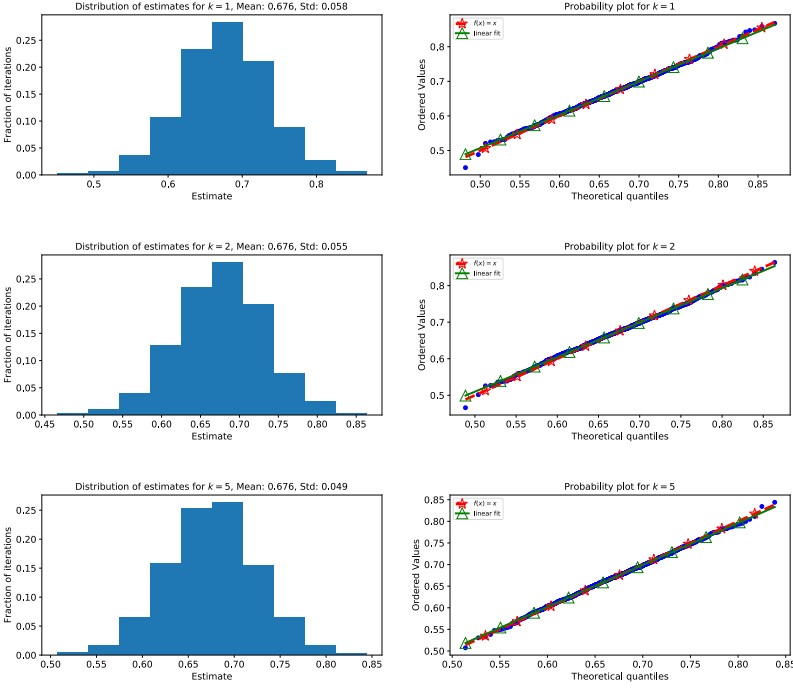

Figure 1: Left: distribution of estimates over 1000 Monte Carlo runs for $k = 1, 2, 5$. Right: the quantile-quantile plot comparison with theoretical asymptotic normal distribution of estimates stemming from our characterization. Means are $0.676, 0.676, 0.676$ and standard deviations are $0.058, 0.055, 0.049$, respectively. $n = 20000$, $D = 20$, $d = 2$, $\mathbf{E}[Y|X] = \frac{1}{1+\exp\{-3X[0]\}}$, $\sigma = 1$. Test point: $x[0] \approx 0.245$, $\mathbf{E}[Y|X = x] \approx 0.676$.

is the vector of covariates and $Y_i$ is the outcome. In Appendix 6, we briefly discuss how our results can be extended to the setting where nuisance parameters and treatments are included in the model.

Suppose that the covariates space $\mathcal{X} \subset \mathbb{R}^D$ is contained in a ball with unknown diameter $\Delta_{\mathcal{X}}$. Denote the marginal distribution of $X$ by $\mu$ and the empirical distribution of $X$ on $n$ sample points by $\mu_n$. Let $B(x, r) = \left\{ z \in \mathbb{R}^D : \|x - z\|_2 < r \right\}$ be the $\ell_2$-ball centered at $x$ with radius $r$ and denote the standard basis for $\mathbb{R}^p$ by $\{e_1, e_2, \ldots, e_p\}$.

Let $\psi : \mathcal{Z} \times \mathbb{R}^p \to \mathbb{R}^p$ be a score function that maps observation $Z$ and parameter $\theta \in \mathbb{R}^p$ to a $p$-dimensional score $\psi(Z; \theta)$. For $x \in \mathcal{X}$ and $\theta \in \mathbb{R}^p$ define the expected score as $m(x; \theta) = \mathbf{E}[\psi(Z; \theta) \mid X = x]$. The goal is to estimate the quantity $\theta(x)$ via local moment condition, i.e.

$$\theta(x) \text{ solves: } m(x; \theta) = \mathbf{E}[\psi(Z; \theta) \mid X = x] = 0.$$

### 2.1. Sub-Sampled Kernel Estimation

**Base Kernel Learner.** Our learner $\mathcal{L}_k$ takes a data set $S$ containing $m$ observations as input and a realization of internal randomness $\omega$, and outputs a kernel weighting function $K_\omega : \mathcal{X} \times \mathcal{X} \times \mathcal{Z}^m \to [0, 1]$. In particular, given any target feature $x$ and the set $S$, the weight of each observation $Z_i$ in $S$ with feature vector $X_i$ is $K_\omega(x, X_i, S)$. Define the weighted score on a set $S$ with internal randomness $\omega$ as $\Psi_S(x; \theta) = \sum_{i \in S} K_\omega(x, X_i, S)\psi(Z_i; \theta)$. When it is clear from context we will

omit $\omega$ from our notation for succinctness and essentially treat $K$ as a random function. For the rest of the paper, we are going to use notations $\alpha_{S,\omega}(X_i) = K_\omega(x, X_i, S)$ interchangeably.

**Averaging over $B$ sub-samples of size $s$.** Suppose that we consider $B$ random and independent draws from all $\binom{n}{s}$ possible subsets of size $s$ and internal randomness variables $\omega$ and look at their average. Index these draws by $b = 1, 2, \ldots, B$ where $S_b$ contains samples in $b$th draw and $\omega_b$ is the corresponding draw of internal randomness. We can define the weighted score as

$$\Psi(x;\theta) = \frac{1}{B} \sum_{b=1}^{B} \Psi_{S_b,\omega_b}(x;\theta) = \frac{1}{B} \sum_{b=1}^{B} \sum_{i \in S_b} \alpha_{S_b,\omega_b}(X_i)\psi(Z_i;\theta). \tag{3}$$

**Estimating $\theta(x)$.** We estimate $\theta(x)$ as a vanishing point of $\Psi(x;\theta)$. Letting $\hat{\theta}$ be this point, then $\Psi(x;\hat{\theta}) = \frac{1}{B} \sum_{b=1}^{B} \sum_{i=1}^{n} \alpha_{S_b,\omega_b}(X_i)\psi(Z_i;\hat{\theta}) = 0$. This procedure is explained in Algorithm 1.

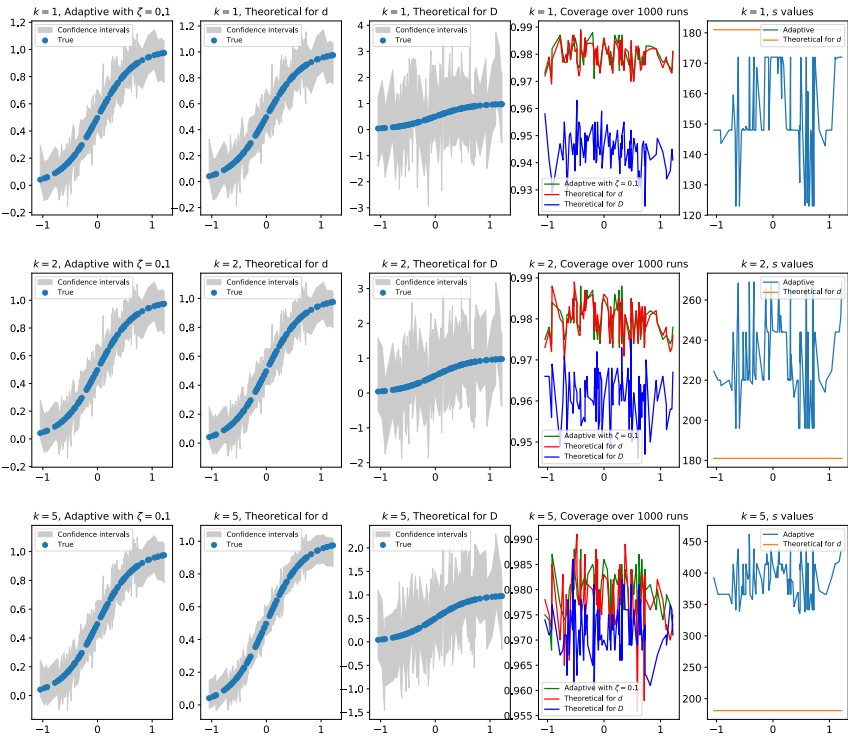

Figure 2: Confidence interval and true values for 100 randomly sampled test points on a single run for $k = 1, 2, 5$ and when (1) left: $s = s_\zeta$ is selected via Proposition 17 with $\zeta = 0.1$, (2) second from the left: $s = n^{1.05d/(d+2)}$, and (3) middle: $s = n^{1.05D/(D+2)}$. Second from the right: coverage over 1000 runs for methods considered. Right: average value of $s_\zeta$ selected via Proposition 17 for $\zeta = 0.1$ for different test points compared to the theoretical value $s = n^{1.05d/(d+2)}$. Here $n = 20000$, $D = 20$, $d = 2$, $\mathbf{E}[Y|X] = \frac{1}{1+\exp\{-3X[0]\}}$, $\sigma = 1$. Nominal coverage: 0.98.

## 2.2. Sub-Sampled $k$-NN Estimation

We especially focus on the case that the weights are distributed across the $k$-NN of $x$. In other words, given a data set $S$, the weights are given according to $K_\omega(x, X_i, S) = 1\{X_i \in H_k(x, S)\}/k$, where $H_k(x, S)$ are $k$-NN of $x$ in the set $S$. The pseudo-code for this can be found in Algorithm 2.

**Complete $U$-statistic.** The expression in Equation (3) is an incomplete $U$-statistic. Complete $U$-statistic is obtained if we allow each subset of size $s$ from $n$ samples to be included in the model exactly once. In other words, this is achieved if $B = \binom{n}{s}$, all subsets $S_1, S_2, \ldots, S_B$ are distinct, and we also take expectation over the internal randomness $\omega$. Denoting this by $\Psi_0(x; \theta)$, we have

$$\Psi_0(x; \theta) = \binom{n}{s}^{-1} \sum_{S \in [n]:|S|=s} \mathbf{E}_\omega \left[ \sum_{i \in S} \alpha_{S,\omega}(X_i) \psi(Z_i; \theta) \right]. \tag{4}$$

Note in the case of $k$-NN estimator we can also represent $\Psi_0$ in terms of order statistics, i.e., $\Psi_0$ is an $L$-statistics (see e.g., Serfling (2009)). By sorting samples in $\mathbf{X} = \{X_1, X_2, \ldots, X_n\}$ based on their distance with $x$ as $\|X_{(1)} - x\| \leq \|X_{(2)} - x\| \leq \cdots \leq \|X_{(n)} - x\|$, we can write $\Psi_0(x; \theta) = \sum_{i=1}^n \alpha(X_{(i)}) \psi(Z_{(i)}; \theta)$ where the weights are given by

$$\alpha(X_{(i)}) = \begin{cases} \frac{1}{k} \binom{n}{s}^{-1} \binom{n-i}{s-1} & \text{if } i \leq k \\ \frac{1}{k} \binom{n}{s}^{-1} \sum_{j=0}^{k-1} \binom{i-1}{j} \binom{n-i}{s-1-j} & \text{if } i \geq k+1. \end{cases}$$

## 2.3. Local intrinsic dimension

We are interested in settings that the distribution of $X$ has some low dimensional structure on a ball around the target point $x$. The following notions are adapted from Kpotufe (2011), which we present here for completeness.

**Definition 1** *The marginal $\mu$ is called* **doubling measure** *if there exists a constant $C_{db} > 0$ such that for any $x \in \mathcal{X}$ and any $r > 0$ we have $\mu(B(x, r)) \leq C_{db}\mu(B(x, r/2))$.*

An equivalent definition of this notion is that, the measure $\mu$ is doubling measure if there exist $C, d > 0$ such that for any $x \in \mathcal{X}, r > 0$, and $\theta \in (0, 1)$ we have $\mu(B(x, r)) \leq C\theta^{-d}\mu(B(x, \theta r))$.

One example is given by Lebesgue measure on the Euclidean space $\mathbb{R}^d$, where for any $r > 0, \theta \in (0, 1)$ we have $\text{vol}(B(x, \theta r)) = \text{vol}(B(x, r))\theta^d$. Building upon this, let $\mathcal{X} \in \mathbb{R}^D$ be a subset of $d$-dimensional hyperplane and suppose that for any ball $B(x, r)$ in $\mathcal{X}$ we have $\text{vol}(B(x, r) \cap \mathcal{X}) = \Theta(r^d)$. If $\mu$ is almost uniform, then we also have $\mu(B(x, \theta r))/\mu(B(x, r)) = \Theta(\theta^d)$.

Unfortunately, this global notion of doubling measure is restrictive and most probability measures are globally complex. Rather, once restricted to local neighborhoods, they become lower dimensional and intrinsically less complex. The following definition captures this intuition better.

**Definition 2** *Fix $x \in \mathcal{X}$ and $r > 0$. The marginal $\mu$ is $(C, d)$-**homogeneous** on $B(x, r)$ if for any $\theta \in (0, 1)$ we have $\mu(B(x, r)) \leq C\theta^{-d}\mu(B(x, \theta r))$.*

Intuitively, this definition requires the marginal $\mu$ to have a local support that is intrinsically $d$-dimensional. This definition covers low-dimensional manifolds, mixture distributions, $d$-sparse data, and also any combination of these examples. These examples are explained in Appendix B.

| **Algorithm 1** Sub-Sampled Kernel Estimation | **Algorithm 2** Sub-Sampled $k$-NN Estimation |
|---|---|
| 1: **Input.** Data $\{Z_i = (X_i, Y_i)\}_{i=1}^n$, moment $\psi$, kernel $K$, sub-sampling size $s$, number of iterations $B$ | 1: **Input.** Data $\{Z_i = (X_i, Y_i)\}_{i=1}^n$, moment $\psi$, sub-sampling size $s$, number of iterations $B$, number of neighbors $k$ |
| 2: **Initialize.** $\alpha(X_i) = 0, 1 \leq i \leq n$ **for** $b \leftarrow 1, B$ **do** | 2: **Initialize.** $\alpha(X_i) \leftarrow 0, 1 \leq i \leq n$ **for** $b \leftarrow 1, B$ **do** |
| 3:     **end**
    **Sub-sampling.** Draw set $S_b$ by sampling $s$ points from $Z_1, Z_2, \ldots, Z_n$ without replacement. | 3:     **end**
    **Sub-sampling.** Draw set $S_b$ by sampling $s$ points from $Z_1, Z_2, \ldots, Z_n$ without replacement |
| 4: **Weight Updates.** $\alpha(X_i) \leftarrow \alpha(X_i) + K_{\omega_b}(x, X_i, S_b)$ | 4: **Weight Updates.** $\alpha(X_i) \leftarrow \alpha(X_i) + 1\{X_i \in H_k(x, S_b)\}/k$ |
| 5: | 5: |
| 6: **Weight Normalization.** $\alpha(X_i) \leftarrow \alpha(X_i)/B$ | 6: **Weight Normalization.** $\alpha(X_i) \leftarrow \alpha(X_i)/B$ |
| 7: **Estimation.** Denote $\hat{\theta}$ as a solution of $\Psi(x; \theta) = \sum_{i=1}^n \alpha(X_i)\psi(Z_i; \theta) = 0$ | 7: **Estimation.** Denote $\hat{\theta}$ as a solution of $\Psi(x; \theta) = \sum_{i=1}^n \alpha(X_i)\psi(Z_i; \theta) = 0$ |

## 3. Assumptions

For non-parametric sub-sampled estimators, the bias and asymptotic variance are tightly connected to the kernel shrinkage and incrementality, formally defined below.

**Definition 3 (Kernel Shrinkage in Expectation)** *The function $\epsilon(s)$ defines a kernel shrinkage in expectation if given a set $S$ containing $s$ i.i.d. observations drawn from distribution $\mathcal{D}$, it satisfies*

$$\epsilon(s) := \mathbf{E}\left[\sup\left\{\|x - X_i\|_2 : K(x, X_i, S) > 0\right\}\right]. \tag{5}$$

**Definition 4 (Kernel Shrinkage in Probability)** *The function $\epsilon(s, \delta)$ defines a kernel shrinkage in probability if given a set $S$ containing $s$ i.i.d. observations drawn from distribution $\mathcal{D}$ w.p. $1 - \delta$ it satisfies*

$$\sup\left\{\|x - X_i\|_2 : K(x, X_i, S) > 0\right\} \leq \epsilon(s, \delta). \tag{6}$$

**Definition 5 (Incrementality of Kernel)** *The incrementality of kernel $K$ when provided with $s$ i.i.d. observations from distribution $\mathcal{D}$ is defined as*

$$\eta(s) = \mathbf{E}\left[\mathbf{E}\left[K(x, X_i, S)|X_i\right]^2\right]. \tag{7}$$

As shown in Wager and Athey (2018), for trees that satisfy some regularity condition, $\epsilon(s) \leq s^{-c/D}$ for a constant $c$. We are interested in shrinkage rates that scale as $s^{-c/d}$, where $d$ is the local intrinsic dimension of $\mu$ on $B(x, r)$. Similar to Oprescu et al. (2018); Athey et al. (2019), we rely on the following assumptions on the moment and score functions.

**Assumption 1** *The moment and score functions satisfy the following:*

1. *The moment $m(x; \theta)$ corresponds to the gradient w.r.t. $\theta$ of a $\lambda$-strongly convex loss $L(x; \theta)$. This also means that the Jacobian $M_0 = \nabla_\theta m(x; \theta(x))$ has minimum eigenvalue at least $\lambda$.*
2. *For any fixed parameters $\theta$, $m(x; \theta)$ is a $L_m$-Lipschitz function in $x$ for some constant $L_m$.*
3. *There exists a bound $\psi_{\max}$ such that for any observation $z$ and any $\theta$, $\|\psi(z; \theta)\|_\infty \leq \psi_{\max}$.*
4. *The bracketing number $N_{[]}(\mathcal{F}, \epsilon, L_2)$ of the function class: $\mathcal{F} = \{\psi(\cdot; \theta) : \theta \in \Theta\}$, satisfies $\log(N_{[]}(\mathcal{F}, \epsilon, L_2)) = O(1/\epsilon)$.*

**Assumption 2** *The moment and score functions satisfy the following:*

1. *For any coordinate $j$ of the moment vector $m$, the Hessian $H_j(x; \theta) = \nabla^2_{\theta\theta} m_j(x; \theta)$ has eigenvalues bounded above by a constant $L_H$ for all $\theta$.*
2. *Maximum eigenvalue of $M_0$ is upper bounded by $L_J$.*
3. *Second moment of $\psi(x; \theta)$ defined as $\mathrm{Var}\left(\psi(Z; \theta) \mid X = x\right)$ is $L_{mm}$-Lipschitz in $x$, i.e.,*

$$\| \mathrm{Var}\left(\psi(Z; \theta) \mid X = x\right) - \mathrm{Var}\left(\psi(Z; \theta) \mid X = x'\right) \|_F \leq L_{mm}\|x - x'\|_2 \,.$$

4. *Variogram is Lipschitz: $\sup_{x \in \mathcal{X}} \| \mathrm{Var}(\psi(Z; \theta) - \psi(Z; \theta') \mid X = x)\|_F \leq L_\psi \|\theta - \theta'\|_2$.*

The condition on variogram always holds for a $\psi$ that is Lipschitz in $\theta$. This larger class of functions $\psi$ allows estimation in more general settings such as $\alpha$-quantile regression that involves a $\psi$ which is non-Lipschitz in $\theta$. Similar to Athey and Imbens (2016); Athey et al. (2019), we require kernel $K$ to be *honest* and *symmetric*.

**Assumption 3** *The kernel $K$, built using samples $\{Z_1, Z_2, \ldots, Z_s\}$, is **honest** if the weight of sample $i$ given by $K(x, X_i, \{Z_j\}_{j=1}^s)$ is independent of $Y_j$ conditional on $X_j$ for any $j \in [s]$.*

**Assumption 4** *The kernel $K$, built using samples $\{Z_1, Z_2, \ldots, Z_s\}$, is **symmetric** if for any permutation $\pi : [s] \to [s]$, the distribution of $K(x, X_i, \{Z_j\}_{j=1}^s)$ and $K(x, X_{\pi(i)}, \{Z_{\pi(j)}\}_{j=1}^s)$ are equal. In other words, the kernel weighting distribution remains unchanged under permutations.*

For a deterministic kernel $K$, the above condition implies that $K(x, X_i, \{Z_j\}_{j=1}^s) = K(x, X_i, \{Z_{\pi(j)}\}_{j=1}^s)$, for any $i \in [s]$. In the next section, we provide general estimation and inference results for a general kernel based on the its shrinkage and incrementality rates.

## 4. Guarantees for sub-sampled kernel estimators

Our first result establishes estimation rates, both in expectation and high probability, for kernels based on their shrinkage rates. The proof of this theorem is deferred to Appendix D.

**Theorem 6 (Finite Sample Estimation Rate)** *Let Assumptions 1 and 3 hold. Suppose that Algorithm 1 is executed with $B \geq n/s$. If the base kernel $K$ satisfies kernel shrinkage in expectation, with rate $\epsilon(s)$, then w.p. $1 - \delta$*

$$\|\hat{\theta} - \theta(x)\|_2 \leq \frac{2}{\lambda} \left( L_m \epsilon(s) + O\left( \psi_{\max} \sqrt{\frac{p\, s}{n} \left( \log\log(n/s) + \log(p/\delta) \right)} \right) \right) . \tag{8}$$

*Moreover,*

$$\sqrt{\mathbf{E}\left[ \|\hat{\theta} - \theta(x)\|_2^2 \right]} \leq \frac{2}{\lambda} \left( L_m \epsilon(s) + O\left( \psi_{\max} \sqrt{\frac{p\, s}{n} \log\log(p\, n/s)} \right) \right) . \tag{9}$$

The next result establishes asymptotic normality of sub-sampled kernel estimators. In particular, it provides coordinate-wise asymptotic normality of our estimate $\hat{\theta}$ around its true underlying value $\theta(x)$. For this result, in addition to the shrinkage, we require the incrementality of the kernel to satisfy some conditions. The proof of this theorem is deferred to Appendix E.

**Theorem 7 (Asymptotic Normality)** *Let Assumptions 1, 2, 3, and 4 hold. Suppose that Algorithm 1 is executed with $B \geq (n/s)^{5/4}$ and the base kernel $K$ satisfies kernel shrinkage, with rate $\epsilon(s, \delta)$ in probability and $\epsilon(s)$ in expectation. Let $\eta(s)$ be the incrementality of kernel $K$ defined in Equation (7) and $s$ grow at a rate such that $s \to \infty$, $n\eta(s) \to \infty$, and $\epsilon(s, \eta(s)^2) \to 0$. Consider any fixed coefficient $\beta \in \mathbb{R}^p$ with $\|\beta\| \leq 1$ and define the variance as*

$$
\sigma_{n,\beta}^2(x) = \frac{s^2}{n} \operatorname{Var}\left[ \mathbf{E}\left[ \sum_{i=1}^s K(x, X_i, \{Z_j\}_{j=1}^s) \left\langle \beta, M_0^{-1}\psi(Z_i; \theta(x)) \right\rangle \mid Z_1 \right] \right].
$$

*Then it holds that $\sigma_{n,\beta}(x) = \Omega\left( s\sqrt{\eta(s)/n} \right)$. Moreover, suppose that*

$$
\max\left( \epsilon(s), \epsilon(s)^{1/4} \left( \frac{s}{n} \log\log(n/s) \right)^{1/2}, \left( \frac{s}{n} \log\log(n/s) \right)^{5/8} \right) = o(\sigma_{n,\beta}(x)). \tag{10}
$$

*Then,*

$$
\frac{\left\langle \beta, \hat{\theta} - \theta(x) \right\rangle}{\sigma_{n,\beta}(x)} \to_d \mathsf{N}(0,1).
$$

Theorems 6 and 7 generalize existing estimation and asymptotic normality results of Athey et al. (2019); Wager and Athey (2018); Fan et al. (2018) to an arbitrary kernel that satisfies appropriate shrinkage and incrementality rates (see Remark 24 in Appendix E). The following lemma relates these two and provides a lower bound on the incrementality in terms of kernel shrinkage. The proof uses the Paley-Zygmund inequality and is left to Appendix F.

**Lemma 8** *For any symmetric kernel $K$ (Assumption 4) and for any $\delta \in [0, 1]$:*

$$
\eta(s) = \mathbf{E}\left[ \mathbf{E}\left[ K(x, X_1, \{Z_j\}_{j=1}^s) \mid X_1 \right]^2 \right] \geq \frac{(1-\delta)^2 (1/s)^2}{\inf_{\rho>0} \left( \mu(B(x, \epsilon(s, \rho))) + \rho \, s/\delta \right)}.
$$

*Thus if $\mu(B(x, \epsilon(s, 1/(2s^2)))) = O(\log(s)/s)$, then picking $\rho = 1/(2s^2)$ and $\delta = 1/2$ implies that $\mathbf{E}[\mathbf{E}[K(x, X_1, \{Z_j\}_{j=1}^s) | X_1]^2] = \Omega(1/s\log(s))$.*

**Corollary 9** *If $\epsilon(s, \delta) = O((\log(1/\delta)/s)^{1/d})$ and $\mu$ satisfies a two-sided version of the doubling measure property on $B(x, r)$, defined in Definition 2, i.e., $c\theta^d \mu(B(x, r)) \leq \mu(B(x, \theta r)) \leq C\theta^d \mu(B(x, r))$ for any $\theta \in (0, 1)$. Then, $\mathbf{E}[\mathbf{E}[K(x, X_1, \{Z_j\}_{j=1}^s) | X_1]^2] = \Omega(1/(s\log(s)))$.*

Even without this extra assumption, we can still characterize the incrementality rate of the $k$-NN estimator, as we observe in the next section.

## 5. Main theorem: adaptivity of $k$-NN estimator

In this section, we provide estimation guarantees and asymptotic normality of the $k$-NN estimator by using Theorems 6 and 7. We first establish shrinkage and incrementality rates for this kernel.

### 5.1. Estimation guarantees for the $k$-NN estimator

In this section we provide shrinkage results for the $k$-NN kernel. As observed in Theorem 6, shrinkage rates are sufficient for bounding the estimation error. The shrinkage result that we present in the following would only depend on the local intrinsic dimension of $\mu$ on $B(x, r)$.

**Lemma 10 (High probability shrinkage for the $k$-NN kernel)** *Suppose that the measure $\mu$ is $(C, d)$-homogeneous on $B(x, r)$. Then, for any $\delta$ satisfying $2\exp\left(-\mu(B(x, r))s/(8C)\right) \leq \delta \leq \frac{1}{2}\exp(-k/2)$, w.p. at least $1 - \delta$ we have $\|x - X_{(k)}\|_2 \leq \epsilon_k(s, \delta) = O\left(\frac{\log(1/\delta)}{s}\right)^{1/d}$.*

We can turn this result into a shrinkage rate in expectation as follows. In fact, by the very convenient choice of $\delta = s^{-1/d}$ combined with the fact that $\mathcal{X}$ has diameter $\Delta_{\mathcal{X}}$, we can establish $O\left((\log(s)/s)^{1/d}\right)$ rate on expected kernel shrinkage. However, a more careful analysis would help us to remove the $\log(s)$ dependency in the bound and is stated in the following corollary:

**Corollary 11 (Expected shrinkage for the $k$-NN kernel)** *Suppose that the conditions of Lemma 10 hold. Let $k$ be a constant and $\epsilon_k(s)$ be the expected shrinkage for the $k$-NN kernel. Then, for any $s$ larger than some constant we have $\epsilon_k(s) = \mathbf{E}\left[\|x - X_{(k)}\|_2\right] = O\left(\frac{1}{s}\right)^{1/d}$.*

We are now ready to state our estimation result for the $k$-NN kernel, which is honest and symmetric. Therefore, we can substitute the expected shrinkage rate established in Corollary 11 in Theorem 6 to derive estimation rates for this kernel.

**Theorem 12 (Estimation Guarantees for the $k$-NN Kernel)** *Suppose that $\mu$ is $(C, d)$-homogeneous on $B(x, r)$, Assumption 1 holds and that Algorithm 2 is executed with $B \geq n/s$. Then, w.p. $1 - \delta$:*

$$\|\hat{\theta} - \theta(x)\|_2 \leq \frac{2}{\lambda}\left(O\left(s^{-1/d}\right) + O\left(\psi_{\max}\sqrt{\frac{p\,s}{n}\left(\log\log(n/s) + \log(p/\delta)\right)}\right)\right), \quad (11)$$

*and*

$$\sqrt{\mathbf{E}\left[\|\hat{\theta} - \theta(x)\|_2^2\right]} \leq \frac{2}{\lambda}\left(O\left(s^{-1/d}\right) + O\left(\psi_{\max}\sqrt{\frac{s\,p\,\log\log(p\,n/s)}{n}}\right)\right). \quad (12)$$

*By picking $s = \Theta\left(n^{d/(d+2)}\right)$ and $B = \Omega\left(n^{2/(d+2)}\right)$ we get $\sqrt{\mathbf{E}\left[\|\hat{\theta} - \theta(x)\|_2^2\right]} = \tilde{O}\left(n^{-1/(d+2)}\right)$.*

### 5.2. Asymptotic normality of the $k$-NN estimator

In this section we prove asymptotic normality of $k$-NN estimator. We first provide a bound on the incrementality of the $k$-NN kernel.

**Lemma 13 ($k$-NN Incrementality)** *Let $K$ be the k-NN kernel and let $\eta_k(s)$ denote the incrementality rate of this kernel. Then, the following holds:*

$$\eta_k(s) = \mathbf{E}\left[\mathbf{E}\left[K(x, X_1, \{Z_j\}_{j=1}^s) \mid X_1\right]^2\right] = \frac{1}{(2s-1)\,k^2}\left(\sum_{t=0}^{2k-2}\frac{a_t}{b_t}\right),$$

*where sequences $\{a_t\}_{t=0}^{2k-2}$ and $\{b_t\}_{t=0}^{2k-2}$ are defined as*

$$a_t = \sum_{i=\max\{0, t-(k-1)\}}^{\min\{t, k-1\}}\binom{s-1}{i}\binom{s-1}{t-i} \quad and \quad b_t = \sum_{i=0}^{t}\binom{s-1}{i}\binom{s-1}{t-i}$$

We can substitute $\eta_k(s)$ in Theorem 7 to prove asymptotic normality of the $k$-NN estimator. The following theorem takes a step further and derives the asymptotic variance of this estimator $\sigma_{n,j}(x)$.

**Theorem 14 (Asymptotic Variance of $k$-NN)** *Let $j \in [p]$ be one of coordinates. Suppose that $k$ is constant while $s \to \infty$. Then, for the $k$-NN kernel*

$$\sigma_{n,j}^2(x) = \frac{s^2}{n} \frac{\sigma_j^2(x)}{k^2 (2s-1)} \zeta_k + o(s/n), \tag{13}$$

*where $\sigma_j^2(x) = \mathrm{Var}\left[ \langle e_j, M_0^{-1}\psi(Z;\theta(x)) \rangle \mid X = x \right]$ and $\zeta_k = k + \sum_{t=k}^{2k-2} 2^{-t} \sum_{i=t-k+1}^{k-1} \binom{t}{i}$.*

Combining results of Theorem 7, Theorem 14, Corollary 11, and Lemma 13 we have:

**Theorem 15 (Asymptotic Normality of $k$-NN Estimator)** *Suppose that $\mu$ is $(C, d)$-homogeneous on $B(x, r)$. Let Assumptions 1, 2 hold and suppose that Algorithm 2 is executed with $B \geq (n/s)^{5/4}$ iterations. Suppose that $s$ grows at a rate such that $s \to \infty$, $n/s \to \infty$, and also $s^{-1/d}(n/s)^{1/2} \to 0$. Let $j \in [p]$ be one of coordinates and $\sigma_{n,j}^2(x)$ be defined in Equation (13). Then,*

$$\frac{\hat{\theta}_j(x) - \theta_j(x)}{\sigma_{n,j}(x)} \to \mathsf{N}(0, 1).$$

*Finally, if $s = n^\beta$ and $B \geq n^{\frac{5}{4}(1-\beta)}$ with $\beta \in (d/(d+2), 1)$. Then, $\frac{\hat{\theta}_j(x) - \theta_j(x)}{\sigma_{n,j}(x)} \to \mathsf{N}(0, 1)$.*

**Plug-in confidence intervals.** Observe that the Theorem 14 implies that if we define $\tilde{\sigma}_{n,j}^2(x) = \frac{s^2}{n} \frac{\sigma_j^2(x)}{2s-1} \frac{\zeta_k}{k^2}$ as the leading term in the variance, then $\frac{\sigma_{n,j}^2(x)}{\tilde{\sigma}_{n,j}^2(x)} \to_p 1$. Thus, due to Slutsky's theorem

$$\frac{\hat{\theta}_j - \theta_j}{\tilde{\sigma}_{n,j}^2(x)} = \frac{\hat{\theta}_j - \theta_j}{\sigma_{n,j}^2(x)} \frac{\sigma_{n,j}^2(x)}{\tilde{\sigma}_{n,j}^2(x)} \to_d \mathsf{N}(0, 1). \tag{14}$$

Hence, we have a closed form solution to the variance in our asymptotic normality theorem. If we have an estimate $\hat{\sigma}_j^2(x)$ of the variance of the conditional moment around $x$, then we can build plug-in confidence intervals based on the normal distribution with variance $\frac{s^2}{n} \frac{\hat{\sigma}_j^2(x)}{2s-1} \frac{\zeta_k}{k^2}$. Note that $\zeta_k$ can be calculated easily for desired values of $k$. For instance, we have $\zeta_1 = 1, \zeta_2 = \frac{5}{2}$, and $\zeta_3 = \frac{33}{8}$ and for $k = 1, 2, 3$ the asymptotic variance becomes $\frac{s^2}{n} \frac{\hat{\sigma}_j^2(x)}{2s-1}, \frac{5}{8} \frac{s^2}{n} \frac{\hat{\sigma}_j^2(x)}{2s-1}$, and $\frac{11}{24} \frac{s^2}{n} \frac{\hat{\sigma}_j^2(x)}{2s-1}$ respectively.

### 5.3. Selecting $s$ adaptively

According to Theorem 12, $s = \Theta(n^{d/(d+2)})$ would trade-off between bias and variance terms. Also, according to Theorem 15, picking $s = n^\beta$ with $d/(d+2) < \beta < 1$ would result in asymptotic normality of the estimator. However, both choices depend on the unknown intrinsic dimension of $\mu$ on the ball $B(x, r)$. Inspired by Kpotufe (2011), we explain a data-driven way for estimating $s$.

Suppose that $\delta > 0$ is given. Let $C_{n,p,\delta} = 2\log(2pn/\delta)$ and pick $\Delta \geq \Delta_{\mathcal{X}}$. For any $k \leq s \leq n$, let $H(s)$ be the $U$-statistic estimator for $\epsilon(s)$ defined as $H(s) = \sum_{S \in [n]:|S|=s} \max_{X_i \in H_k(x,S)} \|x - X_i\|_2 / \binom{n}{s}$. Each term in the summation computes the distance of $x$ to its $k$-nearest neighbor on $S$ and $H(s)$ is the average of these numbers over all $\binom{n}{s}$ possible subsets $S$ (see Remark 34 in Appendix G regarding to efficient computation of $H(s)$). Define $G_\delta(s) = \Delta\sqrt{C_{n,p,\delta}ps/n}$. Iterate

over $s = n, \cdots, k$. Let $s_2$ be the smallest $s$ for which we have $H(s) > 2G_\delta(s)$ and let $s_1 = s_2 + 1$. Note that $\epsilon_k(s)$ is decreasing in $s$ and $G_\delta(s)$ is increasing in $s$. Therefore, there exists a unique $1 \leq s^* \leq n$ such that $\epsilon_k(s^*) \leq G_\delta(s^*)$ and $\epsilon_k(s^* - 1) > G_\delta(s^* - 1)$. We have following results.

**Proposition 16 (Adaptive Estimation)** *Let Assumptions of Theorem 12 hold. Suppose that $s_1$ is the output of the above process. Let $s_* = 9s_1 + 1$ and suppose that Algorithm 2 is executed with $s = s_*$ and $B \geq n/s_*$. Then w.p. at least $1 - 2\delta$ we have $\|\hat{\theta} - \theta(x)\|_2 = O(G_\delta(s^*)) = O\left(\left(\frac{n}{p \log(2pn/\delta)}\right)^{-1/(d+2)}\right)$. Further, for $\delta = 1/n$ we have $\sqrt{\mathbf{E}\left[\|\hat{\theta} - \theta(x)\|_2^2\right]} = \tilde{O}\left(n^{-1/(d+2)}\right)$.*

**Proposition 17 (Adaptive Asymptotic Normality)** *Let Assumptions of Theorem 15 hold. Let $s_1$ be the output of the above process when $\delta = 1/n$ and $s_* = 9s_1 + 1$. For any $\zeta \in (0, (\log(n) - \log(s_1) - \log\log^2(n))/\log(n)))$ define $s_\zeta = s_* n^\zeta$. Suppose that Algorithm 2 is executed with $s = s_\zeta$ and $B \geq (n/s_\zeta)^{5/4}$, then for any coordinate $j \in [p]$, we have $\frac{\hat{\theta}_j(x) - \theta_j(x)}{\sigma_{n,j}(x)} \to \mathsf{N}(0, 1)$.*

## 6. Nuisance parameters and heterogeneous treatment effects

Using the techniques of Oprescu et al. (2018), our work easily extends to the case where the moments depend on, potentially infinite dimensional, nuisance components $h_0$, that also need to be estimated, i.e.,

$$\theta(x) \text{ solves: } m(x; \theta, h_0) = \mathbf{E}[\psi(Z; \theta, h_0) \mid x] = 0. \tag{15}$$

If the moment $m$ is orthogonal with respect to $h$ and assuming that $h_0$ can be estimated on a separate sample with a conditional MSE rate of

$$\mathbf{E}[(\hat{h}(z) - h_0(z))^2 | X = x] = o_p(\epsilon(s) + \sqrt{s/n}), \tag{16}$$

then using the techniques of Oprescu et al. (2018), we can argue that both our finite sample estimation rate and our asymptotic normality rate, remain unchanged, as the estimation error only impacts lower order terms. This extension allows us to capture settings like heterogeneous treatment effects, where the treatment model also needs to be estimated when using the orthogonal moment as

$$\psi(z; \theta, h_0) = (y - q_0(x, w) - \theta(t - p_0(x, w)))(t - p_0(x, w)), \tag{17}$$

where $y$ is the outcome of interest, $t$ is a treatment, $x, w$ are confounding variables, $q_0(x, w) = \mathbf{E}[Y | X = x, W = w]$ and $p_0(x, w) = E[T | X = x, W = w]$. The latter two nuisance functions can be estimated via separate non-parametric regressions. In particular, if we assume that these functions are sparse linear in $w$, i.e.:

$$q_0(x, w) = \langle \beta(x), w \rangle, \qquad\qquad p_0(x, w) = \langle \gamma(x), w \rangle. \tag{18}$$

Then we can achieve a conditional mean-squared-error rate of the required order by using the kernel lasso estimator of Oprescu et al. (2018), where the kernel is the sub-sampled $k$-NN kernel, assuming the sparsity does not grow fast with $n$.

## Conclusion

In this work we studied non-parametric inference when solving general conditional moment equations in high-dimensions and provided estimation and inference guarantees that only depend on the local intrinsic dimension of the covariate space. We confirmed our theoretical findings via numerical simulations.

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
