# OpenReview forum: "Non-parametric Inference Adaptive to Intrinsic Dimension"
_cclear.cc/CLeaR/2022/Conference — CLeaR 2022 Oral_

### Official Review · Reviewer_HtcP · 2021-11-22

**Confidence:** 2
**Overall Score:** 6

**Main Review:**

## Technical quality
The paper establishes the rate and asymptotic normality of the estimator. In particular, under an unknown intrinsic dimension, a data-driven procedure is shown to be adaptive.

In terms of its technical strengths and weaknesses in the context of the literature, I am unable to provide an informed assessment as I am not familiar with related works.

## Clarity
The paper seems quite dense and I find it very hard to follow. After all, there is not much room left after introducing the definitions / conditions and stating the results.

There are a few broken sentences in the first paragraph of page 3.

## Originality
The authors seem to be able to build upon previous works and generalize their results, e.g., the definition of "intrinsic dimension" and asymptotic normality of the sub-sampled k-NN estimator.

## Significance
As authors mentioned in the Introduction, I presume that the results here are relevant to several problems in causal inference. But the impact is not immediately clear to me. Perhaps the authors can comment on the application of their results.

**Summary:**

The paper considers estimators defined as solution to a conditional moment equation when the conditioned covariates are of high dimension but low "intrinsic dimension". Estimator based on a sub-sampled k-NN weighted estimating equation is considered and its properties are studied.

---

> ### Author Response · Authors · 2021-12-04
> **Response to Reviewer HtcP**
>
> We would like to thank the reviewer for their insightful review. Below we address your questions:
>
> 1. The paper seems quite dense and I find it very hard to follow. After all, there is not much room left after introducing the definitions/conditions and stating the results. There are a few broken sentences in the first paragraph of page 3.
> **Response.** We will fix the broken sentences in the first paragraph of page 3 and try to edit some of the text to make it less dense.
>
> 2. As the authors mentioned in the Introduction, I presume that the results here are relevant to several problems in causal inference. But the impact is not immediately clear to me. Perhaps the authors can comment on the application of their results.
>
> **Response.** Our paper is the first to show the possibility of achieving non-parametric inference for generalized moment equations with rates depending only on intrinsic dimension. This to some extent, can justify the success of non-parametric methods in causal inference. Our paper is mostly there to explain WHY existing methods work, rather than suggest new methods.
> As we noted in the first paragraph, most forms of regression are special cases of Generalized Method of Moments, and find wide application in economics, statistics, public policy, political science, bio-statistics etc...

---

### Official Review · Reviewer_QJkx · 2021-11-22

**Confidence:** 3
**Overall Score:** 8

**Main Review:**


The reviewer is familiar with conditional moment models and their application, familiar with intrinsic dimensions, but only have textbook knowledge about non-parametric estimation.

Here are my detailed comments:
1)	The problem can use more motivation, e.g. in which application area is it more common to have high dimensional set of conditioning variables? In many cases, one can estimate the causal structure and given the estimated causal structure, only a smaller number of variables need to be conditioned on.
2)	Simulations: Is there any reason that the specific function family/data generation function was chosen? It would also be interesting to see numerical illustration/demonstration of the bound.
3)	It would be helpful if the author can comment of the complexity of the estimation.
4)	Assumptions: Can the authors comment of some of the assumptions with respect to testability or how likely they are going to be met? In other words, guidance for when the methodology is applicable for people who are consider to use it on real world data.


**Summary:**

The paper provided an algorithm for estimating conditional moments based on a sub-sampled ensemble of the kNN Z estimator, and provided bound for estimation error. Theorems were also provided for adaptively choosing hyper parameter s to achieve the desired estimates. The reviewer find the paper clear and informative.

---

> ### Author Response · Authors · 2021-12-04
> **Response to Reviewer QJkx**
>
> We would like to thank the reviewer for their insightful review. Below we address your questions:
>
> 1. We provide some examples of high dimensional spaces with low intrinsic dimensions in Appendix B. As mentioned there, one example is when the conditioning variable is an image, i.e., we want to do inference based on images. You can find other examples there too. You are correct that in many cases, one can estimate causal effects only using a small number of conditioning variables.
>
> 2. Please see Appendix C where we explain the data generating process. We would be happy to add numerical illustrations of our theoretical bounds for some simple examples.
>
> 3. The complexity of estimation can be decomposed into two parts: (1) calculating the weights using $B$ different sub-samples of size $s$ for different points and (2) solving the weighted generalized moment equation. The first part is usually not that difficult and complexity is $O(B C_{\text{$k$-NN}})$ where $C_{\text{$k$-NN}}$ is the complexity of $k$-NN algorithm executed on $s$ samples (this is generally $O(sd)$ unless approximate algorithms are used or further structure is assumed). The difficulty for solving the weighted generalized moment equation depends on the structure of this function. For example, in the case of convex moment, this can be solved very efficiently.
>
> 4. Our assumptions stated in Assumptions 1 and 2 are mainly conditions on the score and moment function that need to be estimated. These assumptions are satisfied for many well-known moment functions used in practice (regression, quantile regression, etc.). Our assumption on the intrinsic dimension corresponds to having a high-dimensional feature space which is less complex locally. For the examples of feature spaces with small intrinsic dimension, see Appendix B. Our paper basically provides general estimation and inference guarantees for any combination of these two.

---

### Official Review · Reviewer_uVEy · 2021-11-23

**Confidence:** 5
**Overall Score:** 7

**Main Review:**

Nonparametric estimation of conditional moment models in high dimensions has been widely and well studied in the literature. In recent years, many works have been dedicated to establishing the large sample properties for the case where the conditioning variable has a small intrinsic dimension and found that the optimal convergence rate depends only on $d$; see the references in the paper and see also Jiao et al. (2021) and the references therein. The K-NN method of approximating the geodesic distance on the lower intrinsic dimensional manifold is commonly used (see Tenenbaum et al., 2000; Kpotufe, 2011; Chen and Muller, 2012 among others). The authors generalise the asymptotic results of Wager and Athey (2018), Athey et al. (2019) for sub-sampled random forests and that of Fan et al. (2019) for sub-sampled 1-NN estimator in the high dimension setting to their sub-sampled K-NN generalised method of moments in the low intrinsic dimension regime. Inspired by the ideas in Kpotufe (2011), the authors propose a data-driven method to choose the tuning parameter.

There are two main contributions of the paper: (1) the explicit form of the asymptotic variance of the estimator and (2) the data-driven tuning parameter that can guarantee the optimal convergence rate of the estimator. Nevertheless, I have the following major concerns:

1. The motivation for using the sub-sampling and fixing the value of $k$ of the K-NN is unclear. Without the sub-sampling, the estimator is essentially a uniform kernel estimator, where the kernel shrinkage acts like the bandwidth of a standard kernel estimator and depends on the sample size $n$ and the K-NN parameter $k$. However, it is known that if $k$ satisfies certain rate conditions, such an estimator can also achieve the optimal convergence rate of a nonparametric estimator, and one may also incorporate kernel functions other than a uniform one to the K-NN to improve the practical performance; see Bickel and Li (2007); Kpotufe (2011); Padilla et al. (2020) for example. Then, what is the advantage of using the sub-sampling?
2. The authors treat the K-NN parameter $k$ as a constant in the paper. As I mentioned in the first question, the kernel shrinkage parameter should depend on $k$. What would the asymptotic results of the estimator be if we allow $k$ to depend on the sample size $n$?
3. In practice, $k$ has to be chosen to construct the estimator. Could the authors discuss how to determine $k$ and how does $k$ affect the practical performance?
4. In the theorems in the paper, are $\lambda$, $\psi_{\max}$ and $p$ constants lie in $(0,\infty)$? If so, taking $B=1$ and $s=n$, which corresponds to the K-NN estimator without sub-sampling, in the upper bound of the square root of the mean squared error of the estimator, we have $O\\{\lambda^{-1}\psi_{\max}\sqrt{n^{-1}ps\log\log(pn/s)}\\}=O(1)$. This may not converge to 0. Why is that? Please correct me if I misunderstand anything.
5. In the paper, the authors provide the explicit form of the asymptotic variance and say that if they have an estimate of the variance, they can build plug-in confidence intervals based on the asymptotic normality result. However, the exact estimator of the variance is not stated. Since this is one of the main contributions of the paper, the authors should be clear here. Based on the formula of the variance, many different estimation methods can be used. Do they all have similar theoretical and practical performance?
6. In the simulation, the authors consider only one setting: one model, one sample size, one number of dimensions of the covariates and one intrinsic dimension. This does not reflect the theories well.
    (a) The sample size is set to $n=20000$ while the intrinsic dimension $d$ is only 2. Usually, with $d=2$, a sample size of $n=5000$ should be large enough to obtain a good performance. The authors may consider $n=1000,2000$ and $5000$ to present the convergence of the estimation and the coverage of the confidence intervals.
    (b) It would also be interesting to see a comparison between the proposed sub-sampled method and the K-NN estimator without sub-sampling but with $k$ chosen by cross-validation.
7. Although the authors mention that the conditional moment models are related to heterogeneous treatment effect problems, the connection of the results to causal learning and reasoning is not straightforward. Indeed, in the treatment effect problems, other issues need to be considered in the estimation, such as confounding. Therefore, the estimation and inference results in the paper cannot be directly applied. This concerns if the paper is suitable for the conference.

**Reference**

Athey, S., Tibshirani, J. and Wager, S. (2019) Generalized random forests. *The Annals of Statistics*, **47**, 1148-1178.

Bickel, P. J. and Li, B. (2007) Local polynomial regression on unknown manifolds. *Complex datasets and inverse problems, Institute of Mathematical Statistics*, 177-186.

Chen, D. and Muller, H.-G. (2012) Nonlinear manifold representations for functional data. *The Annals of Statistics*, **40**, 1-29.

Fan, Y., Lv, J. and Wang, J. (2018) Dnn: A two-scale distributional tale of heterogeneous treatment effect inference. *arXiv preprint arXiv:1808.08469*.

Jiao, Y., Shen, G., Lin, Y. and Huang, J. (2021) Deep nonparametric regression on approximately low-dimensional manifolds. *arXiv preprint arXiv:2104.06708*.

Kpotufe, S. (2011) K-NN regression adapts to local intrinsic dimension. *In Advances in Neural Information Processing Systems*, 729-737.

Tenenbaum, J. B., de Silva, V. and Langford, J. C. (2000) A global geometric framework for nonlinear dimensionality reduction. *Science*, **290**, 2319-2323.

Padilla, O. H. M., Sharpnack, J. and Chen, Y. (2020) Adaptive nonparametric regression with the K-nearest neighbour fused lasso. *Biometrika*, **107**, 293-310.

**Summary:**

This paper considers a special case of nonparametric generalised method of moments in high dimensions, where the conditioning variable has a small intrinsic dimension. In particular, the authors propose an estimator that solves a locally weighted empirical conditional moment equation with sub-sampled kernel or sub-sampled K-NN weights. The authors show that the estimator can achieve the optimal convergence rate depending only on the small intrinsic dimension but not the high dimension where the covariates belong to, which escapes the well-known curse of dimensionality. Furthermore, the authors provide an explicit form of the asymptotic variance of their estimator, which helps construct the confidence intervals. The paper also proposes an adaptive data-driven approach for choosing the sub-sample size and proves that the estimator with the selected sub-sample size achieves the optimal rate.

---

> ### Author Response · Authors · 2021-12-04
> **Response to Reviewer uVEy**
>
> We would like to thank the reviewer for their insightful review. Below we address your questions:
>
> 1. It is correct that sub-sampling is not required for achieving optimal convergence rate in nonparametric setting. However, sub-sampling is required to make the $k$-NN a regular asymptotic normal estimator for inference. Without sub-sampling this estimator can achieve asymptotic normality but it would be irregular. For the same reason, Fan et. al. (2019) and Athey et. al. (2019) considered subsampled kernel estimators to achieve asymptotic normality, while Bickel and Li (2007), Kpotufe (2011) and Padilla et al. (2020) that study $k$-NN without subsampling do not address inference.
>
> 2. You are right that the kernel shrinkage depends on both values of $k$ and $s$ (See Lemma 10). As stated in the paper, our results in Theorems 6 and 7 are valid for all sub-sampled kernel estimators, including $k$-NNs with $k$ depending on the sample size $n$. Once the conditions required for kernel shrinkage and incrementality rates for the $k$-NN established in Lemma 10 and 13 are satisfied, all the results in Theorems 12, 14, and 15 remain valid. For example, our convenient choice of $\delta=s^{-1/d}$ requires that $s^{-1/d} \leq \frac{1}{2}\exp(-k/2)$. One can incorporate these conditions and allow for settings where $k$ can get larger as $n$ gets larger. However, to keep the presentation simpler, and also keeping the adaptive parameter tuning problem simpler (tuning one parameter instead of two), we decided to mainly focus on the case where $k$ is constant.
>
> 3. As mentioned in the answer to Q2, we decided to keep $k$ fixed and provided data-drive adaptive way for tuning $s$. These two may be related, but sub-sampling allows for inference and that is our main focus in this paper.
>
> 4. Our main focus is on situations where $s << n$, where sub-sampling allows inference. In this case, we need to control for the error that is induced by sampling. This would contribute an additional term of order $\tilde{O}(s/n)$ which as you mentioned does not exist in the case where $s=n$.
>
> 5. This is a great question and it can be interesting follow-up work. However, studying different estimators for estimating the variance and their performances is outside of the scope of this work.
>
> 6. We would be happy to provide results with smaller values of sample size $n$ in the final version.
>
> 7. The conditional moment equation formulation in (1) allows for approaches that can handle situations with confounding.  For example, instrumental variables is a case where the conditional moment restriction might hold for some set of exogenous (i.e. unconfounded) instruments $X$, even though the function $\psi$ takes as input data $Z$ that may suffer from confounding.
> With that said, even if this paper only accounted for the case with observed confounders, we would be in good company: Athey et. al. (2019) and Fan et. al. (2019) both considered cases with observed confounding variables as well.

---

> > ### Comment · Reviewer_uVEy · 2021-12-26
> > **Response to the authors**
> >
> > Thank you for answering my questions. Your responses resolve all of my main concerns, and I can now update the score to 7: Good paper, accept.

---

### Decision · Program_Chairs · 2022-01-12

**Decision:**

Accept (Oral)

**Comment:**

The paper is concerned with non-parametric estimation and focuses on explaining why non-parametric estimation happen to work well even in high dimensions. It shows that the estimation and asymptotic normality results only depend on the intrinsic dimension  of the data.

The proposed approach is based on a sub-sampled ensemble of the k-nearest neighbors and an adaptive procedure is proposed to identify the intrinsic dimension . It might be interesting to comment the relationship among this procedure and the paper: Estimating the intrinsic dimension of datasets by a minimal neighborhood information, Facco et al., Nature 2017.

All reviewers are interested in the presented approach, in the line of and extending previous works by Athey et al., and the authors' rebuttal clearly addressed the questions.

Some efforts in making the revised version of the paper easier to follow are mandatory: please discuss the motivations, complexity of solving the weighted generalized moment equation depending on its structure, and provide some guidelines for practitioners.

Some ask the authors to make the paper easier to follow.